# Comparative Analysis of Multilayer Lead Oxide-Based X-ray Detector Prototypes

**DOI:** 10.3390/s22165998

**Published:** 2022-08-11

**Authors:** Emma Pineau, Oleksandr Grynko, Tristen Thibault, Alexander Alexandrov, Attila Csík, Sándor Kökényesi, Alla Reznik

**Affiliations:** 1Physics Department, Lakehead University, Thunder Bay, ON P7B 5E1, Canada; 2Institute for Nuclear Research, H-4026 Debrecen, Hungary; 3Department of Electrical and Electronic Engineering, University of Debrecen, H-4026 Debrecen, Hungary; 4Thunder Bay Regional Health Research Institute, Thunder Bay, ON P7B 6V4, Canada

**Keywords:** lead oxide, PbO, photoconductor, direct conversion, X-ray detector, signal lag, sensitivity

## Abstract

Lead oxide (PbO) photoconductors are proposed as X-ray-to-charge transducers for the next generation of direct conversion digital X-ray detectors. Optimized PbO-based detectors have potential for utilization in high-energy and dynamic applications of medical X-ray imaging. Two polymorphs of PbO have been considered so far for imaging applications: polycrystalline lead oxide (poly-PbO) and amorphous lead oxide (a-PbO). Here, we provide the comparative analysis of two PbO-based single-pixel X-ray detector prototypes: one prototype employs only a layer of a-PbO as the photoconductor while the other has a combination of a-PbO and poly-PbO, forming a photoconductive bilayer structure of the same overall thickness as in the first prototype. We characterize the performance of these prototypes in terms of electron–hole creation energy (*W_±_*) and signal lag—major properties that define a material’s suitability for low-dose real-time imaging. The results demonstrate that both X-ray photoconductive structures have an adequate temporal response suitable for real-time X-ray imaging, combined with high intrinsic sensitivity. These results are discussed in the context of structural and morphological properties of PbO to better understand the preparation–fabrication–property relationships of this material.

## 1. Introduction

The demand for advanced radiation medical imaging technologies sustains research interest in novel X-ray photoconductive materials and structures for direct conversion imaging detectors. Such technology would allow for enhancing the sensitivity of X-ray detectors in a way that improves the radiation safety of diagnostic and image-guided procedures. In direct-conversion flat panel X-ray imagers (FPXIs), a uniform layer of the photoconductor is deposited over large-area readout electronics. The photoconductor acts as an X-ray-to-charge transducer, i.e., it absorbs X-rays and directly creates electron–hole pairs, which are subsequentially separated by an applied electric field to generate a signal. Stabilized amorphous selenium (a-Se) is currently the sole commercially viable X-ray photoconductor used in FPXIs [1,2]. A-Se is especially suitable for mammographic detectors since a relatively thin layer (~200 μm) is sufficient to absorb nearly all of the soft X-rays in a relatively low-energy range of mammography while providing excellent charge collection efficiency [3]. For higher energy applications, such as radiography and fluoroscopy, a-Se-based detectors would have to utilize a much thicker (more than 1 mm) layer of a photoconductor. However, growing such thick layers is not only technologically challenging due to the emerging problem of the adhesion of thick layers to the substrate, but the charge collection efficiency of a thick photoconductor is also compromised. Therefore, currently, a-Se-based X-ray detectors for the diagnostic energy range are not used commercially, and alternative materials and technologies must be considered.

X-rays cannot be efficiently converged yet, so detectors must have a large active area. The research and development of large-area materials with good X-ray stopping power are focused on disordered (amorphous and polycrystalline) phases of high-Z (atomic number) X-ray photoconductors. The application necessitates reproducible technology that also allows uniform layers to be deposited directly on the imaging electronics at temperatures that the electronics can withstand. Materials of interest include BiI_3_ [4], PbI_2_ [5,6,7], HgI_2_ [7,8,9], ZnO [10], CdTe [11], Cd_1_-_x_Zn_x_Te [12], and PbO [13].

It should be noted that this list of “traditional” materials for use in direct-conversion medical imaging detectors has recently been supplemented by a new class of materials, i.e., perovskites [14]. Perovskite semiconductors have emerged as some of the most promising materials for optoelectronic applications, including solar cells, light-emitting diodes, and lasers [14,15,16]. Unlike most other materials for photonics, perovskites contain high-Z elements, which makes them effective X-ray-to-charge convertors in the diagnostic energy range. Although very promising, perovskites are still at an early stage of development and require a solution for their poor temporal performance [15,16].

Among the “traditional” materials listed above, lead oxide stands out since its polycrystalline phase (poly-PbO) has a history of application in Plumbicon video pick-up tubes for commercial optical imaging. Its proven suitability for broadcast, fluoroscopy, and angiography with X-ray image intensifiers makes it a particularly promising material to expand the direct conversion scheme of X-ray medical imaging to general radiography and fluoroscopy [17]. The advantage of poly-PbO is its greater X-ray sensitivity compared to that of a-Se. Its *W_±_*, the average energy required to generate a single detectable (i.e., collected) electron and hole pair (ehp), is less than half that of commercial a-Se X-ray photoconductors at the same operational electric field of 10 V/µm [18]. This is clinically useful as poly-PbO-based detectors could be dose-efficient to an exposure level that is half of what is used for a-Se detectors. The desire for low-exposure operation is in accordance with the principle of “as low as reasonably achievable” (ALARA). Unfortunately, the first detector prototype with a thick (~300 µm, suitable for radiography [19]) layer of poly-PbO photoconductor exhibited poor temporal performance [13], with large signal lag following X-ray exposure. This inhibits its potential application in dynamic imaging, i.e., fluoroscopy, which is the most clinically demanding procedure. 

The shortcomings of poly-PbO were alleviated by the development of a new polymorph—amorphous lead oxide (a-PbO) [20]. A-PbO shows a significant improvement in temporal performance and dark current suppression from its polycrystalline counterpart [21,22,23,24]. However, this improvement comes at a cost: the amorphization of the a-PbO layer requires the deposition to be performed with the aid of low-energy oxygen ion bombardment (the so-called ion-assisted thermal deposition process), which adds to the cost and technological complexity of detector manufacturing [25]. 

The use of multilayer photoconductive structures is a standard solution in direct conversion detectors. Different approaches to a-Se multilayered structures have been considered in the development of practical a-Se direct conversion detectors. Decades of research and commercialization have led to patented device structures comprised of p-type layers to block electron injection (a-Se + 1–38% As [26], a-As_2_Se_3_ [27,28], a-Se doped with Cl [29], Sb_2_S_3_ [1], organic, inorganic, and polycarbonate films [30]); n-type layers to introduce hole trapping (alkali doped a-Se [26,27,28,29,31], a-Se doped with oxide or halogenide [26], organic, and inorganic films [30]); and crystallization prevention layers (a-Se doped with 10–33% of As, Sb, or Bi [30], a-Se with As, S, Te, P, Sb, or Ge, As_2_S_3_ [32]). Overall, the current solutions utilize a multilayer structure consisting of a thick photoconductive layer of stabilized a-Se sandwiched between one or two adjacent blocking layers needed to maintain an acceptable dark current [3,33]. The a-Se acts as an X-ray-to-charge transducer while blocking layers prevent (block) carrier injection from electrodes. The development of the a-Se blocking structures allows for the application of a much higher electric field to the a-Se photoconductor without an increase in the dark current, thus improving the schubweg i.e., the average distance drifted before a carrier is lost to traps (the schubweg is a product of the carrier mobility *µ*, the lifetime *τ*, and the field *F*) and avoiding depth-dependent charge collection. This in turn resulted in significantly improved detective quantum efficiency (DQE), signal-to-noise ratio (SNR), and suppressed lag [1]. Out of all the investigated blocking structures, those employing a thin polyimide (PI) blocking layer sandwiched between the electrode and the photoconductor are particularly promising given their demonstrated compatibility with both a-Se and a-PbO X-ray-to-charge transducers and the ability to suppress the dark current in a-Se more efficiently than what has been achieved with other blocking layers [22,34]. PI also boasts a relatively simple and cost-effective fabrication process that is compatible with large-area imaging electronics, which is discussed in the next section of this work [34].

Given the efficacy of a-Se multilayer structures, we explored similar ideas to make practical PbO X-ray-to-charge transducers. Our approach was to combine the two polymorphic forms of the same material, namely PbO in a multilayer structure with a PI blocking layer. In this study, we fabricated a single-pixel PI/a-PbO/poly-PbO X-ray detector prototype and performed a comparative analysis of its X-ray photoconductive properties with a single-pixel PI/a-PbO detector prototype [22,23]. Both prototypes (a-PbO and multilayer) have a thin (1 µm) PI blocking layer sandwiched between the bottom electrode and the photoconductor to reduce the dark current and allow for high-field operation. We characterized the performance of both prototypes in terms of the electron–hole pair creation energy, *W_±_*, and signal lag, which determine the intrinsic X-ray sensitivity and the temporal response to X-ray excitation, respectively. We discuss the advantages and disadvantages of each proposed detector prototype, linking them to the morphology of the layers, and offer insight into which is better suited to expand the success of direct conversion detectors over the diagnostic energy range.

## 2. Materials and Methods

### 2.1. Preparation of Detectors 

Both the PI/a-PbO and PI/a-PbO/poly-PbO detector prototypes utilize a 1 µm thick PI layer spin-coated onto an ITO-coated glass substrate (UniversityWafer Inc., Boston, MA, USA), a photoconductor layer deposited on the prepared substrate, and a gold (Au) contact, 1.1 mm in diameter, sputtered on top of the photoconductor. The photoconductor in the first prototype consists of a 19 µm thick a-PbO layer; the second bilayer configuration has a relatively thin (4 μm) layer of a-PbO and a thicker (14 μm) layer of poly-PbO on top. The thickness of a gold layer in the PI/a-PbO detector is 40 nm; however, since the top layer of poly-PbO in the bilayer configuration has higher roughness than a-PbO [25,35], a thicker layer of gold (200 nm) was utilized. These structures are shown schematically in Figure 1.

The PI blocking layer was obtained as a polyamic acid precursor dissolved in an n-methyl-2-pyrrolidone (NMP) based solvent (HD MicroSystems, LLC, Parlin, NJ, USA), spin-coated in an even layer on top of the substrate, and then cured on a hot plate to achieve full imidization of the film. The curing substrate was kept under a gentle flow of dry nitrogen until the curing was complete. Kapton tape was used to mask the corners of the substrate so that they remained uncoated, with the conductive ITO layer exposed, for the later purpose of electrical connection during the sample characterization. The parameters used to coat and cure an approximately 1 μm uniform layer are detailed in Table 1 [22].

The poly-PbO and a-PbO layers were deposited by conventional and ion-assisted thermal deposition techniques, respectively. In conventional thermal evaporation, high-purity PbO powder (Chemsavers, Inc., Bluefield, VA, USA) is evaporated in a metal crucible heated to ~1000 °C in an atmosphere of molecular oxygen. The PbO vapor is deposited onto a substrate and heated to ~100 °C while the substrate is rotating to ensure a uniform thickness of the PbO across the sample [17,25]. In ion-assisted evaporation, the rotating substrate is simultaneously bombarded with low-energy oxygen ions. These oxygen ions originate from an ion source that can be controlled during deposition to achieve the optimal ion current density and ion energy, allowing for the amorphization of the depositing layer. Detailed descriptions of both deposition processes have been previously reported in [25]. Table 2 summarizes the main deposition parameters used for the growth of PbO photoconductors.

### 2.2. Detector Characterization

In this work, the characterization of the detector prototype was based on parameters of interest: X-ray sensitivity (in terms of *W_±_*) and temporal performance (in terms of signal lag) measured with the X-ray-induced photocurrent method (XPM). The X-ray properties are discussed in the context of structural peculiarities of PbO revealed by scanning electron microscopy (SEM) images and the Raman spectra of our prototypes.

Prior to the XPM measurements, the detector rested for a few hours in the dark. As was previously shown [22], the dark current of the PI/a-PbO detector decreased with time after the application of a bias voltage and began to saturate after 15 min due to the redistribution of the electric field over time. Our results suggest that waiting 15 min after the bias application allows sufficient time for the electric field to reach a steady-state condition so that both the dark current and the ehp creation energy are mostly stabilized. The mechanism of this redistribution and the physical reason this occurs are reported in [36]. The biased detector was irradiated with an X-ray beam of given energy and flux, and the induced photocurrent was measured (Figure 2). The X-ray tube (Underwriters Laboratories Inc., Northbrook, IL, USA) had a tungsten anode, a 1.3 mm Al filter (used to attenuate low-energy photons and minimize Compton backscattering noise), and a 2 mm lead collimator (used to form narrow-beam geometry). The exposure was monitored by an ionization chamber (Keithley Instruments, Cleavland, OH, USA).

#### 2.2.1. X-ray Sensitivity

The sensitivity of the X-ray detector refers to its reciprocal value *W_±_*—the average energy required to generate a single detectable (i.e., collected) electron–hole pair (ehp) [1]. *W_±_* is defined as
(1)$$W_{\pm }=\frac{E_{abs}}{N_{ehp}}\;,$$
where *E_abs_* is the total energy from the X-ray irradiation absorbed in the photoconductive layer, and *N_ehp_* is the number of electron–hole pairs collected by the electronics. The radiation energy *E_abs_*, deposited onto the photoconductor, is calculated from the absorbed fraction of the X-ray beam spectrum (i.e., X-ray photon fluence) Φ*_inc_*(*E*) of the X-ray tube for a given tube voltage and measured in the plane of the detector exposure *X* using the following equation [21,23]:(2)$$E_{abs}=A\;X\;\int^{\;}\frac{\mu_{en}\left(E\right)}{\mu \left(E\right)}\Phi_{inc}\left(E\right)\left(1-e^{-\mu \left(E\right)d}\right)dE,\;\;$$
where *A* = 0.95 mm^2^ is the active area of the detector, *µ_en_* and *µ* are the energy absorption and linear attenuation coefficients for the PbO photoconductor derived from [37] using the measured density of each of the photoconductors, and *d* is the photoconductor’s thickness. Φ*_inc_*(*E*) was simulated using the standard Tucker–Barnes–Chakraborty (TBC) model [38] for the given parameters (tungsten anode, 1.3 mm Al filter, tube current of 200 mA, tube voltage of 60 kVp). The number of collected ehps, *N_ehp_*, was obtained by integrating the photocurrent (the measured current minus the dark current). The X-ray sensitivity was measured in a range of electric fields (5–40 V/µm) relevant to the operation of X-ray detectors and at the mean X-ray energy of 37 keV. Theoretically, at infinitely strong fields, *W_±_* should approach a value of *W_±_*° xdefined by the Klein rule for X-ray photo-interaction. The Klein rule provides an approximate relationship between W±° and the bandgap, *E_g_*, of the photoconductor, given by *W_±_*° ≈ 3*E_g_* [39]. The experimental X-ray sensitivity that approaches theoretical values is an indicator of the optimal X-ray-to-charge conversion efficiency.

#### 2.2.2. Temporal Response

To evaluate the temporal performance, a pulsed-mode XPM was utilized [18,21,22,34,40]. This method allows for the investigation of the X-ray response dynamics during the pulsed irradiation and for evaluating the carry-over of a residual signal into the next frames, i.e., lag, at different frame rates under conditions resembling the practical operation of a detector during dynamic imaging. A detector, biased at different fields (3–30 V/µm), was irradiated with a 1-s long X-ray pulse with a mean energy of 37 keV, which was modulated by a rotational chopper at a variable frequency (5–30 Hz), as depicted in Figure 3. The chopper (Stanford Research Systems, Inc., Sunnyvale, CA, USA) was placed as close as possible to the X-ray tube aperture to preserve the quasi-rectangular shape of the X-ray beam that is incident upon the detector.

Signal lag is defined as the residual current flowing through the photoconductive layer after the termination of the X-ray exposure. Quantitatively, the signal lag was calculated as the ratio of the average photocurrent post-irradiation (*PC*_off_–*DC*) (when X-rays are blocked by the chopper), to the average photocurrent during the irradiation (*PC*_on_–*DC*) [22,34,41]:(3)$$Lag=\frac{PC_{off}-DC}{PC_{on}-DC}$$
where the dark current *DC* is measured as the average signal prior to the irradiation.

#### 2.2.3. Morphological Analysis

The cross-sectional morphologies of the a-PbO and bilayer samples were investigated with a Thermo Fisher Scientific-Scios 2 dual-beam scanning electron microscope (FIB SEM, Waltham, MA, USA) operated at a low accelerating voltage (2 keV). The application of a low acceleration voltage increases the yield of secondary electrons generated near the surface [41]. As a result, a setting can be found where the incoming and outgoing currents are the same and the sample current is equal to zero. This means that no electric conductivity of the sample is required to eliminate the charge accumulation and thus, insulating materials can be studied without applying a conductive coating layer, which may modify the morphology of the surface [41].

Raman spectroscopy measurements were performed with a Renishaw inVia Raman spectrometer (Renishaw, Wotten-under-Edge, UK) a resolution of 1 cm^−1^. The Raman spectra were measured on the surface of the sample with a 532 nm laser line and 2400 lp/mm grating. Prior to the measurements, the system was calibrated to a silicon peak at 520 cm^−1^. The laser’s intensity was carefully adjusted to prevent light-induced changes, i.e., crystallization of the film.

## 3. Results

### 3.1. X-ray Sensitivity

Figure 4 shows the experimental *W_±_* calculated using Equations (1) and (2) for both the PI/a-PbO and PI/a-PbO/poly-PbO prototypes measured at a mean X-ray energy of 37 keV, as well as poly-PbO and a-Se-based detector prototypes replotted from [18,42] for comparison, in a range of operational electric fields. In the poly-PbO, the *W_±_* was measured by pulse height spectroscopy (PHS) at an X-ray energy of 60 keV, while in the a-Se, the *W_±_* was measured at 40 keV. The Klein rule [39,43] value (which is about three times that of the bandgap) for PbO, using an approximate bandgap of 1.9 eV, is also plotted for reference. The value of 1.9 eV was chosen as the reference point for consistency with the previously published works; the value of 1.9 eV was reported for the mobility gap of a-PbO [44], and for the bandgap of poly-PbO (within the experimental error) [1,17,44]. Note that the poly-PbO prototype in this comparison does not have a blocking layer. This should not have affected the charge collection efficiency of the detector but its dark current was much higher, preventing accurate measurements of the *W_±_* in the poly-PbO prototypes at fields higher than 10 V/µm. The a-Se prototype in this comparison utilized a standard commercial-grade xeroradiography plate with a thin polymer-blocking layer under the anode. These data are representative of a typical commercial a-Se-based X-ray detector. These two caveats must be kept in consideration for all further comparisons presented in this work. 

As is evident from Figure 4, for all materials under comparison here, the *W_±_* decreased with the electric field. At 10 V/µm (an applied electric field commonly used in practical detector operation), the *W_±_* of ~23 eV/ehp for a-PbO based detector compares favorably with the ~45 eV/ehp for a-Se. This indicates an approximately twice-as-high intrinsic sensitivity of a-PbO than a-Se-based detectors at the same field [3,45]. Meanwhile, the bilayer detector prototype displayed an even better ehp creation energy of 7.8 eV/ehp at 10 V/µm. This suggests that the bilayer PbO detector prototype could be operated at less than one-fifth of the X-ray exposure-per-frame required for a-Se detectors. Furthermore, the *W_±_* approached the theoretical value of 5–7 eV/ehp which, as previously stated, is the marker of the optimal sensitivity of a PbO-based detector. 

### 3.2. Temporal Response

Figure 5 shows a typical response of the PI/a-PbO detector to a 1-s-long X-ray pulse modulated at 30 Hz. The detector exhibited a constant amplitude of photocurrent in each frame, and after the exposure was terminated, the signal rapidly dropped. Such behavior was identical for the studied applied fields of 5–20 V/μm, although, the higher the field, the greater the amplitude of the photo signal. Therefore, qualitatively, the PI/a-PbO detector demonstrated good temporal performance with low lag and no signal build-up.

As for PI/a-PbO/poly-PbO, at 10 V/µm, the temporal performance resembles that of the PI/a-PbO prototype (Figure 6a). However, as is evident from Figure 6b, at 20 V/µm, the response of this detector to the modulated X-ray beam irradiation is different. At fields ≥15 V/µm, the signal gradually built up in each frame, suggesting the presence of an injection current that dynamically increased during the exposure, and then, following exposure, decayed over time, causing signal lag.

A quantitative evaluation of the signal lag is presented in Figure 7 with a direct comparison of the signal lag values calculated using Equation (3) for the detector prototypes at a range of electric fields and modulation frequencies. The amplitudes used for deriving PC_off_, PC_on,_ and DC in Equation (3) are indicated in Figure 6. The data for a-PbO were previously reported in [22], the data for poly-PbO were extracted from [18] and the data for a-Se were extracted from [34,40].

For the a-PbO detector prototype, the lag increased with an increasing modulation frequency (as expected, since the residual current was probed with a shorter delay after the termination of exposure by a chopper blade) and reached a value of 1.9% at 30 Hz and 10 V/µm. However, the temporal performance improved, and the lag decreased with the application of a stronger electric field, down to 0.9% at 30 Hz (the frequency needed for fluoroscopy) at 20 V/µm. Fortunately, this matches the trend of the X-ray sensitivity improving with increasing fields (Figure 4). For the bilayer detector prototype, the field dependence of the lag was not quite as straightforward. At first, the lag decreased with an increasing field (8.25% at 5 V/µm), down to 0.8% at 15 V/µm, but increased at higher fields (4.7% at 30 V/µm). The latter is associated with dynamic X-ray-induced injection [18,41] at this field, as can be seen in Figure 6b, as a build-up of a photocurrent during the irradiation; this is discussed in the next section. It is also important to keep the reproducibility of these results in mind. The low lag values at intermediate fields (10–15 V/µm) can be reported with a high degree of confidence, shown by the small error bars calculated for these measurements. The lag values of both the a-PbO and multilayer PbO prototypes compare very favorably with those for a-Se-based detectors, as shown in Figure 7.

### 3.3. Morphological Analysis

Figure 8 compares the cross-sectional SEM images of the PI/a-PbO structure with the bilayer a-PbO/poly-PbO structure [35] and the poly-PbO sample [25]. As expected, the results differ significantly: while the a-PbO grew as a highly packed layer (Figure 8a), both the poly-PbO grown on a glass substrate and the poly-PbO grown on an a-PbO sublayer were porous and highly inhomogeneous. However, the structure of the poly-PbO layer in the bilayer configuration (Figure 8c) was different from the single poly-PbO layer structure (Figure 8b). Indeed, in the bilayer structure, a thin and uniform layer of amorphous material in the lower part of the sample transitions into the inhomogeneous disordered layer of polycrystalline material comprising small overlapping flakes. In contrast, a single layer of poly-PbO consists of a porous network of individual platelets oriented mainly in the growth direction [25]. These platelets were a few microns in diameter and about 50 nm thick, which is considerably larger than the size of the flakes in the bilayer PbO. This also resulted in a lower porosity and higher bulk density of the bilayer PbO film: ~2 g/cm^3^ (20% of the crystalline PbO density) in the single poly-PbO layer versus ~3.3 g/cm^3^ (35%) in the bilayer PbO.

As can be seen in Figure 8a, the a-PbO phase appears to have maintained uniformity throughout the layer. However, a precise analysis of the cross-section morphology revealed a local violation of the homogeneity of the structure, appearing as crystallites and cracks (see insets of Figure 8a). It should be noted that thin a-PbO layers (i.e., thinner than 10 μm, such as those used in the bilayer PbO prototype) are smooth, uniform, and free of these structural imperfections, as reported previously [25].

## 4. Discussion

The theoretical X-ray sensitivity is governed by the Klein rule for X-ray photoconductors. Intrinsic charge creation energy, *W_±_*°, is directly proportional to the bandgap; it is a material parameter and independent of the experimental setup. The experimental charge creation energy, *W_±_*, on the other hand, is not a material parameter. Rather, it is a characteristic of a detector prototype in a particular experimental setup depending on the applied contact and electric field, X-ray energy, exposure, and temperature. Experimentally, for the majority of X-ray photoconductors, the *W_±_* is higher than the theoretical value [3,18,23,45]. This indicates that not all photogenerated charges are collected by the read-out electronics; rather, they undergo deep trapping or recombination within the photoconductor [23], making the *W_±_* field-dependent, as shown in Figure 4. For all samples under discussion here, the *W_±_* decreased with the field. The field dependences of recombination and deep trapping are described in detail in numerous publications [23,42,45,46,47]; therefore, without going into detail, here, we highlight these concepts within the framework of columnar and Langevin recombination. The probability for an X-ray-generated charge to escape recombination is proportional to the electric field; however, at a certain field, the time required for an electron–hole pair to meet in space becomes electric field-independent and the *W_±_* will saturate [23,45]. In the best-case scenario, the time required for an electron–hole pair to meet in space becomes lower than the timescale of a recombination event (10^−12^ s), and the *W_±_* increases at its fundamentally low value. This saturation value is advantageous from a research and development standpoint, as having a range of fields with high sensitivity gives a higher probability of being able to choose an operational electric at which the sensitivity is high and that also has adequate temporal performance.

As is evident in Figure 4, the bilayer detector prototype demonstrates the lowest *W_±_* when compared to other structures at the same fields. Its *W_±_* goes down with the field reaching its theoretical value, *W_±_*°, at ~15 V/µm. Although the *W_±_* becomes almost field-independent, at high electric fields, there is still a slight decrease in the *W_±_* with the field that eventually makes the experimental curve cross the dotted line that corresponds to the Klein rule value for PbO. The notable injection observed at fields higher than 15 V/µm, previously discussed in the context of the response to modulated irradiation (Figure 6) makes it possible to explain this seemingly strange result. This phenomenon, previously reported in [18], is a type of injection, also known as X-ray photoconductive gain. In contrast to regular injection, it does not affect the dark current of the detector when there is no incident irradiation. Rather, X-ray photoconductive gain arises when a large number of photogenerated carriers within the photoconductor cause a temporary enhancement of the electric field at the photoconductor–electrode interface. We believe that prototypes that have poly-PbO in direct contact with the top Au electrode, i.e., the bilayer sample investigated in this work and the poly-PbO in [18], are more susceptible to this effect due to the rough surface of poly-PbO that may further strengthen the electric field at the tips of the individual platelets (flakes). The X-ray-induced injection current makes XPM measurements overestimate the *W_±_* values (recall that the *W_±_* is calculated from the ratio of absorbed radiation energy to collected charge carriers). We could not easily distinguish between the X-ray-triggered injection current and the photocurrent in our resultant single-pulse XPM waveform, so empirically, it seems that we have collected more charge carriers for the same absorbed energy. In addition, the X-ray photoconductive gain was responsible for the degradation of the temporal response with the application of a stronger electric field: as the injection increased, the photo-signal built up, and signal lag arose.

We believe that this injection current from the top Au electrode was indeed triggered by the X-ray photoconductive gain (rather than by a high electric field) for a few reasons. First of all, for the entire range of the electric field, the dark current (measured in the absence of X-ray radiation) gradually increased with the field; at 15 V/μm and higher, it remained below 0.2 nA, which is lower than the *PC_off_* level in Figure 6b. Secondly, after the application of a bias, the dark current remained steady prior to the X-ray irradiation (as seen at the initial portion of the current transient in Figure 6b). Finally, at the beginning of the irradiation, a photo signal built up and the *PC_off_* values increased in each frame, meaning that the injection current increased only during the exposure. The fact that the lag increased with the modulation frequency at a fixed electric field supports our conclusion that the X-ray photoconductive gain from the electrodes was the major cause for the signal lag in all the materials under investigation in this work. Nonetheless, since our results suggest that at electric fields ≤15 V/µm, there was no noticeable X-ray-triggered injection from the bias electrodes, we confirm that the experimental *W_±_* values at 15 V/µm and below represent the true electron–hole creation energy and its field dependence in this structure. 

For the PI/a-PbO detectors, the saturated *W_±_* value was around 22 eV/ehp and occurred at fields greater than 10 V/µm. This is still greater than both the theoretical sensitivity of 5–7 eV/ehp and the experimental value for a bilayer structure, which indicates that recombination still occurred at a higher rate than in the bilayer detector, but its probability was no longer controlled by the applied electric field [23]. In terms of the temporal response and blocking characteristics, the lag remained low and decreased with the field, while the PI layer controlled the injection current [22]. As is shown in Figure 5, during the exposure to modulated irradiation, the a-PbO detector exhibited a constant amplitude in successive frames. This indicates that the dark current was constant and the photocurrent neither increased (due to X-ray photoconductive gain) nor decreased (due to the space-charge limited transport) [48,49]. The latter suggests that the presence of the PI blocking layer does not affect the charge collection efficiency, acting, as it was designed, to be a blocking layer rather than an insulator.

Despite the fact that the lag problem in the a-PbO layers was solved, our study reveals an important challenge in a-PbO technology at this stage of development, i.e., the degradation of structural homogeneity in thick layers grown by ion-assisted deposition. Indeed, the initial research on a-PbO X-ray photoconductor was performed using comparatively thin (2–10 μm) layers. However, the practical thickness of PbO-based X-ray-to-charge transducers is in the range of 50 μm (for use in breast tomosynthesis) to 600 μm (for use in fluoroscopy) [19]. Even though for the PI/a-PbO prototype tested in this study, the a-PbO layer was only 19 μm thick (thinner than required for practical applications but thicker than what was tested previously [21]), the structure is lacking structural homogeneity typical for thin a-PbO layers, as can be seen from the results of the analysis of the morphology of the cross-sections (Figure 8a). A possible explanation is the local recrystallization of the growing layer that creates inclusions, or micro-crystallites, shown in the inset of Figure 8a. Indeed, a-PbO is fundamentally metastable with respect to its polycrystalline counterpart, but the amorphous PbO phase undergoes gradual recrystallization at temperatures above 200 °C, as was previously shown in [25]. Although an investigation into the possible local phase transitions within a-PbO layers grown by ion-assisted thermal deposition remains for future work, here, Figure 9 supports the hypothesis. The Raman spectrum of the as-deposited thicker (19 µm) a-PbO layer, shown in Figure 8a, has characteristic peaks, with wavenumbers matching those for a thin (7 μm) a-PbO layer annealed at 200 °C. This indicates that the as-grown thick a-PbO layers already started to recrystallize during the deposition. 

At this point, we can only speculate that the energy delivered by the ion bombardment to the growing layer (needed to improve the surface atom mobility and to compress the atoms within the film into a denser amorphous structure) led to self-annealing in certain areas. This effect was not found in thin layers, as their thermal conductivity was sufficient to allow for a cooling effect from a relatively cold substrate. With increasing thickness, the growing layer became less thermally conductive and energy dissipation to the cold substrate was suppressed. This resulted in a lower cooling rate, thus providing the longer time needed to reach the appropriate substrate temperature, causing a localized phase transition, i.e., pinpoint recrystallization of the layer. It seems reasonable to assume that the onset time for self-annealing became shorter as the temperature within the growing layer increased. In addition, micro-crystallites had more time to grow during the prolonged deposition required to grow thick a-PbO layers. Of course, the effects of local strain in promoting recrystallization should be carefully investigated [50,51]. Finally, thermal strain may occur between different parts of a substrate and a layer, causing structural imperfections, including cracks throughout the film [52]. We could link the appearance of structural imperfections with the degradation in sensitivity, i.e., a higher *W_±_* of the PI/a-PbO prototype seen in Figure 4 can be explained by the enhanced recombination at the recombination sites located at many structural imperfections.

The problems with thick layer growth identified above are expected as it is well-known that controlling important properties (i.e., adhesion, stability, stoichiometry, and reproducibility of structural peculiarities) is much more complex in thick layers than in thin films. While we will continue our research effort on making a-PbO technology mature and reproducible, we will also consider an alternative approach, i.e., multilayer detector prototypes that combine a thin bottom a-PbO layer with a much thicker upper poly-PbO layer. The goal of this approach is to combine the fast temporal response and structural stability of thin a-PbO layers with the significant reduction in the technological complexity that would come with utilizing poly-PbO to make up the bulk of the photoconductor thickness. This approach overcomes the problem faced with thick a-PbO photoconductors since the a-PbO layer in a multilayer configuration is relatively thin and no ion bombardment, responsible for self-annealing, is utilized in the poly-PbO layer deposition. In general, at this stage of their development, a-PbO/poly-PbO detector prototypes show immense promise for application in direct-conversion X-ray medical imaging. The achieved *W_±_* at 10 V/μm is ~5 times lower than that of commercial a-Se X-ray photoconductors and ~3 times lower than that of a-PbO prototypes, offering higher X-ray sensitivity and a lower radiation dose required to acquire a radiographic image. The problem of the injection current and the associated signal lag at the electric fields above 15 V/μm can be solved by improving the blocking characteristics, especially at the poly-PbO/Au interface. A second blocking layer placed at the poly-PbO/Au interface that could planarize the rough surface of the poly-PbO layer would prevent the injection current, improving lag and maintaining the excellent X-ray sensitivity of these detectors at higher electric fields. Among the potential materials with a compatible deposition method are cellulose acetate (CA) [53], GeO_2_ [54], and CeO_2_ [55]. Overall, combining the demonstrated performance with a simple, low-cost, large-area compatible deposition process makes the reported a-PbO/poly-PbO X-ray photoconductive structures especially useful in low-dose imaging applications. Achieving a low theoretical value of electron–hole creation energy at ≤10 V/µm (which is a practical electric field) is a great accomplishment in the continuous development of direct-conversion medical imaging technologies. To the best of our knowledge, PbO is the first of the potential disordered X-ray photoconductors to simultaneously demonstrate high sensitivity and adequate temporal performance. This suggests that the employment of these X-ray-to-charge transducers in commercial detectors will make them dose-efficient down to the lowest exposures used in fluoroscopy.

## 5. Conclusions

In this work, a comparative analysis was performed on two different PbO-based single-pixel direct-conversion X-ray detector prototypes, including the evaluation of X-ray sensitivity as an indicator of its suitability for low-dose application and temporal performance, required for image clarity in dynamic multi-frame image procedures. The a-PbO-based detector prototype showed consistently good temporal performance at operational electric fields and modulation frequencies, which is promising for its application in multi-frame imaging procedures. A-PbO layers are deposited using the ion-assisted thermal evaporation technique, and the deposition conditions (i.e., deposition rate, ion energy and flux, substrate temperature, and operational pressure) should be further optimized to provide growth of the uniform layers with practical thicknesses for applications in X-ray detectors (up to 600 μm). 

The multilayer PbO-based detector prototype shows very impressive X-ray sensitivity (a dose six times lower than what is required by a-Se detectors) at a practical field of 10 V/μm. This has the core benefit to the imaging system designer of either reducing the incident X-ray dose to acquire a diagnostic image and thereby improving the overall safety of general radiography and image-guided interventional procedures, or at typical X-ray exposures to improve the image quality. The multilayer detector also demonstrated a suitable temporal performance at practical fields (up to 15 V/μm), although it degraded at higher fields due to X-ray photoconductive gain. The next steps in the development of this prototype will be to explore options for an electron blocking layer made of a semi-insulating polymer, such as cellulose acetate (CA), GeO_2_, or CeO_2_, and deposited between the poly-PbO photoconductor and the top electrode. It is envisioned that such a four-layered photoconductive structure will combine high X-ray sensitivity with low signal lag.

## Figures and Tables

**Figure 1 sensors-22-05998-f001:**
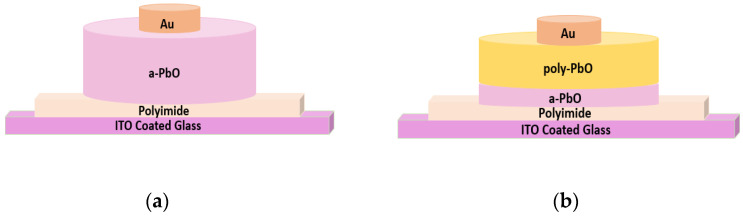
Schematic of (**a**) PI/a-PbO multilayer detector structure, (**b**) PI/a-PbO/poly-PbO multilayer detector structure (schematics not to scale).

**Figure 2 sensors-22-05998-f002:**
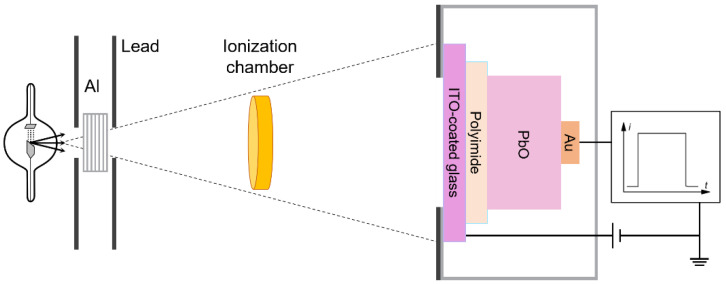
Schematic representation of the X-ray induced photocurrent method (XPM) used in this work for the characterization of the X-ray response performance of detector prototypes (not to scale) [23].

**Figure 3 sensors-22-05998-f003:**
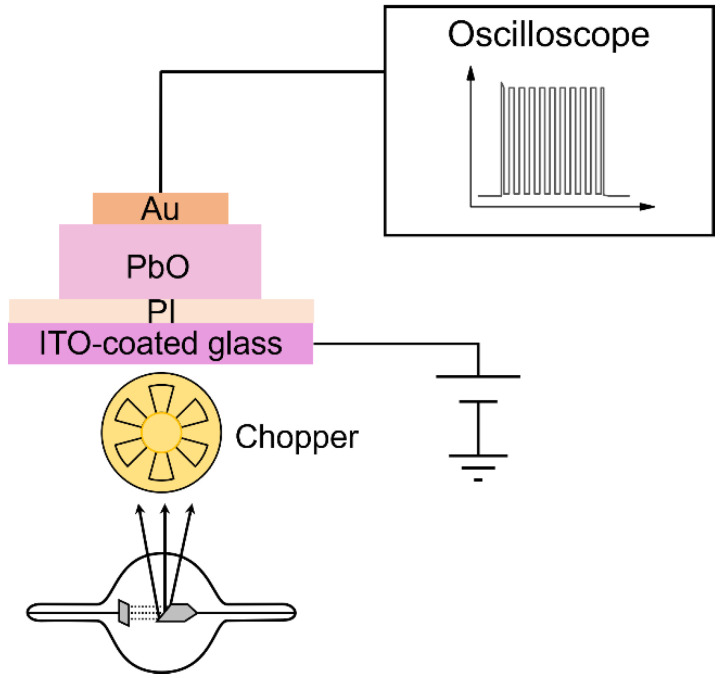
Schematics of pulsed mode XPM setup used to characterize temporal performance (not to scale).

**Figure 4 sensors-22-05998-f004:**
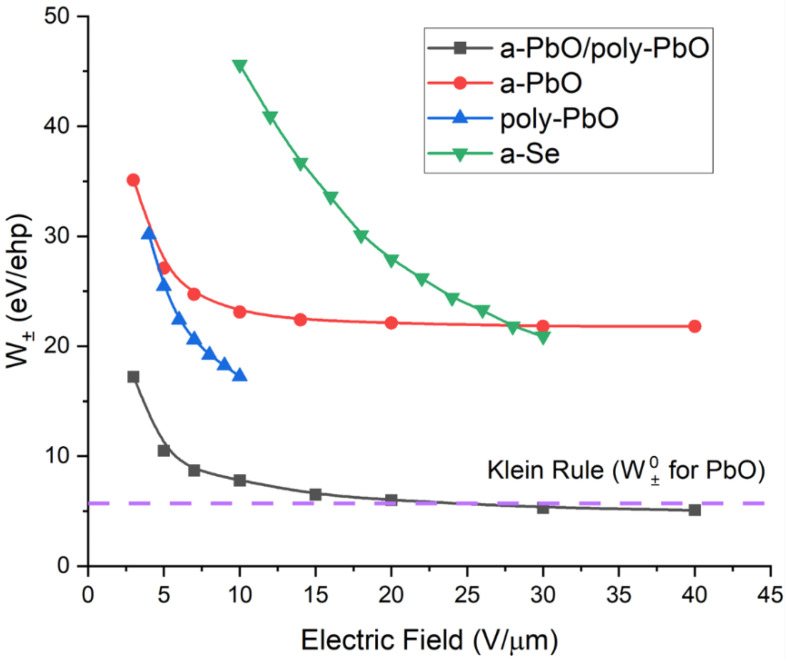
*W_±_* at various operational electric fields for photoconductive materials of interest. Data for a-PbO, poly-PbO, and a-Se were extracted from Refs. [18,22,42], respectively.

**Figure 5 sensors-22-05998-f005:**
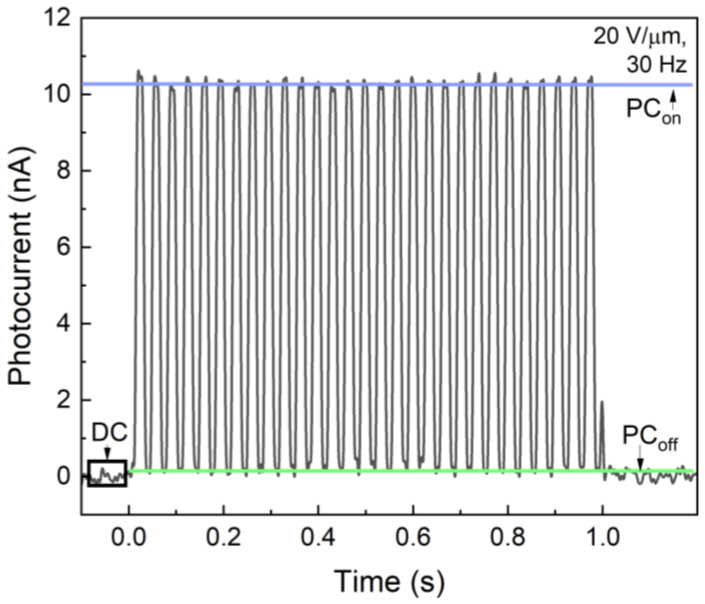
A typical photocurrent of the PI/a-PbO detector due to modulated irradiation.

**Figure 6 sensors-22-05998-f006:**
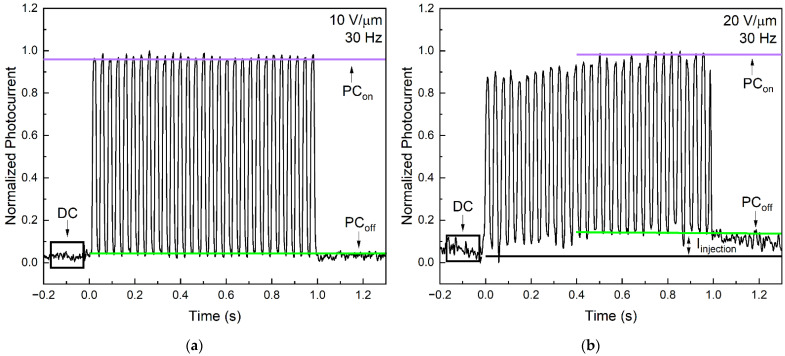
Response of PI/a-PbO/poly-PbO detector prototype to modulated 30 Hz X-ray irradiation at (**a**) 10 V/µm and (**b**) 20 V/µm, showing the build-up of a photocurrent at high fields.

**Figure 7 sensors-22-05998-f007:**
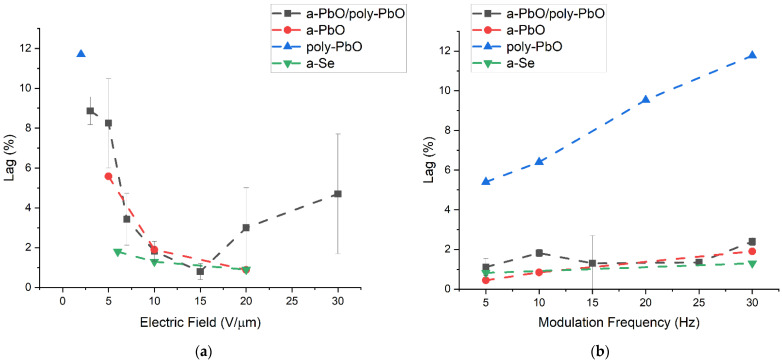
Signal lag as a function of (**a**) electric field for 30 Hz modulation frequency and (**b**) varying frequency for an applied electric field of 10 V/µm in a-PbO, bilayer PbO and a-Se, and 2 V/µm for poly-PbO [18,22,34,40]. Error bars have been added for the a-PbO/poly-PbO sample that was evaluated for this work. Data points are connected with dashed lines as a guide for the eye.

**Figure 8 sensors-22-05998-f008:**
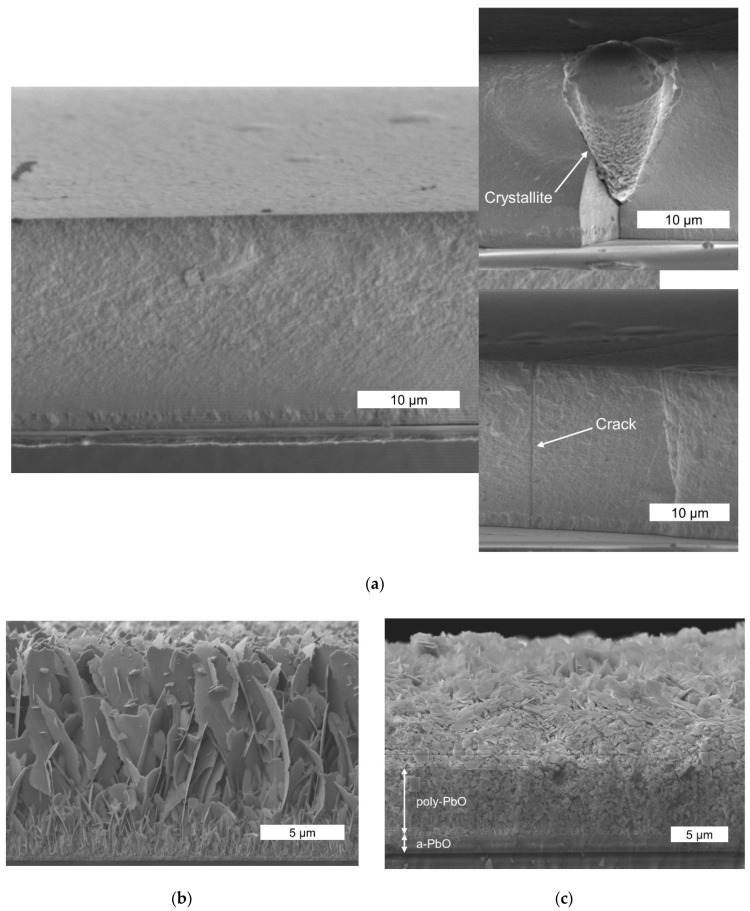
The SEM cross-sectional view of (**a**) a-PbO, (**b**) poly-PbO [25], and (**c**) a-PbO/poly-PbO bilayer structure [35]. The insets of Figure 9a show structural defects that appear in thick a-PbO layers. (**b**) Image adapted from Semeniuk et al. [25] with the permission of Springer Nature. (**c**) Image adapted from Grynko et al. [35] with permission of Springer Nature.

**Figure 9 sensors-22-05998-f009:**
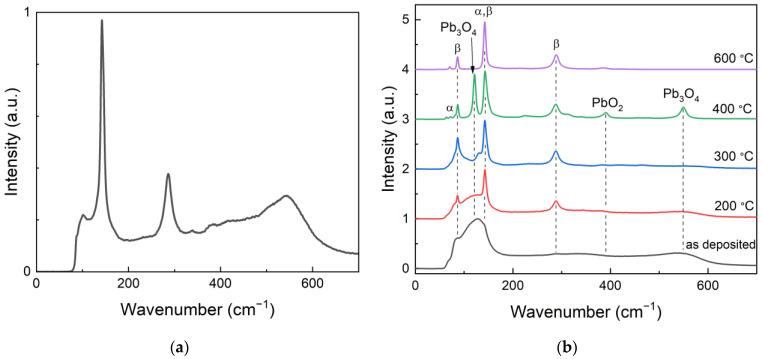
Raman spectra of (**a**) as-deposited thick a-PbO layer discussed in this work and (**b**) thin a-PbO layer annealed at different temperatures. The spectrum of the as-grown thick layer demonstrates the same characteristic features of thin a-PbO annealed at the temperature of 200 °C. (**b**) Image adapted from Semeniuk et al. [25] with the permission of Springer Nature.

**Table 1 sensors-22-05998-t001:** Parameters for spin-coated deposition and curing of PI blocking layer [22].

Spin Coating Parameters				Hot Plate Curing	
	Step 1	Step 2	Step 3	Final Temperature, °C	350
Duration of Step, s	5	30	7	Duration of Curing, min	30
Speed, rpm	500	6000	0	Rate of Change in Temperature, °C/min	4
Acceleration, rpm/s	550	990	−1100	Nitrogen Flow, sL/min	2

**Table 2 sensors-22-05998-t002:** Deposition parameters relevant to the deposition of PbO layers.

Deposition Parameters		
	Conventional Thermal Deposition (poly-PbO)	Ion-Assisted Thermal Deposition (a-PbO)
Base Pressure, Pa	5 × 10^−5^	5 × 10^−5^
Process Pressure, Pa	2 × 10^−1^	6 × 10^−2^
Furnace Temperature, °C	1000	1000
Substrate Temperature, °C	100	100
Deposition Rate, µm/min	1	0.22
Ion Energy, eV		50
Ion Flux, mA/cm^2^		0.2

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
