# Peer review of "Comparative Analysis of Multilayer Lead Oxide-Based X-ray Detector Prototypes"

_sensors, 2022, doi:10.3390/s22165998_

Round 1

Reviewer 1 Report

Comments to authors:

This work compares X-ray detectors made of different lead oxide based structures. The performance is better than commercial a-se detector and the research is of practical interest. This work can be published at sensors with some clarifications addressed.

1)     Introduction section. Perovskites are emerging semiconductor materials for flat panel X-ray imager. More background about Perovskites FPXI should be included and discussed. 

2)     Line 94, a full name of PI should be provided the first time it appears in the paper. 

3)     Line 159-160. Could the authors give some comments or insights on why biasing can reduce and stabilize the dark current?

4)     Line 172-173, is the description of sensitivity definition accurate? from equation (1), Nehp is the collected electron-hole pairs. So “The sensitivity of the x-ray detector refers to its reciprocal value W± – the average energy required to generate a single detectable electron-hole pair (ehp)”, should this sentence be changed to “The sensitivity of the x-ray detector refers to its reciprocal value W± – the average energy required for a single electron-hole pair (ehp) collected”

5)     Line 178-179, “The energy, Eabs, deposited onto the photoconductor is calculated from the absorbed fraction of the simulated spectrum (i.e., x-ray photon fluence) of the x-ray tube for a given tube voltage and measured in the plane of the detector exposure”. This expression seems not straightforward to understand. Could the authors provide more details on how Eabs is obtained? Is it from simulation or measurement? If measured, how it is measured? Reference [17] and [19] might contain relevant info, but it is important to provide more details in this work, as Eabs is critical affecting evaluation of detector performance. 

6)     Line 205-206, Can the authors provide some clarifications on “It was evaluated by irradiating the biased detector with a series of 1-s long x-ray pulses, modulated at a frequency of 5–30 Hz”. It seems confusing. With 1-s long pulses, how can the pulsed modulated at frequency higher than 1 Hz? At 5–30 Hz, wouldn’t the X-ray pulse duration be smaller than 1s?. 

Besides, rigorously, the X-ray pulse generated by chopper is a trapezoidal shape as function of time, instead of a rectangular shape. The X-ray beam size and chopper size affect the slope of the trapezoid. Given that consideration, is it a good approximation the X-ray pulse has a rectangular shape?  If not, is there any influence of the pulse shape on the detector lag characterization? The authors are expected to provide some explanations, comments or references on these questions to justify the setup. 

Reviewer 2 Report

See attached file. 

Reviewer 3 Report

Very interesting work. It is not clear to me that the only disadvantages of poly-PbO are "poor temporal performance [12], with large signal lag following x-ray exposure; this inhibits its potential application in real-time imaging, fluoroscopy, which is the most clinically demanding procedure."  I would expect that scaling the poly-PbO detector over a large area is also quite challenging especially non uniformity in transport at the polycrystalline boundaries.

The idea of using a-PbO which in theory would be more uniform for large area X-ray detectors could potentially solve the large area detector manufacturing challenge as well.  This point should be mentioned in the manuscript somewhere.

Round 2

Reviewer 2 Report

The authors have done an excellent job in responding to comments and revising the manuscript.